# Contrasting effects on deep convective clouds by different types of aerosols

Jonathan H. Jiang[1], Hui Su [1], Lei Huang[1,2], Yuan Wang[3], Steven Massie[4], Bin Zhao[2], Ali Omar [5] & Zhien Wang[4,6,7]

Convective clouds produce a significant proportion of the global precipitation and play an important role in the energy and water cycles. We quantify changes of the convective cloud ice mass-weighted altitude centroid ($Z_{IWC}$) as a function of aerosol optical thickness (AOT). Analyses are conducted in smoke, dust and polluted continental aerosol environments over South America, Central Africa and Southeast Asia, using the latest measurements from the CloudSat and CALIPSO satellites. We find aerosols can inhibit or invigorate convection, depending on aerosol type and concentration. On average, smoke tends to suppress convection and results in lower $Z_{IWC}$ than clean clouds. Polluted continental aerosol tends to invigorate convection and promote higher $Z_{IWC}$. The dust aerosol effects are regionally dependent and their signs differ from place to place. Moreover, we find that the aerosol inhibition or invigoration effects do not vary monotonically with AOT and the variations depend strongly on aerosol type. Our observational findings indicate that aerosol type is one of the key factors in determining the aerosol effects on convective clouds.

[1] Jet Propulsion Laboratory, California Institute of Technology, Pasadena 91109 CA, USA. [2] Joint Institute for Regional Earth System Science and Engineering, University of California, Los Angeles 90095 CA, USA. [3] Division of Geological and Planetary Sciences, California Institute of Technology, Pasadena 91106 CA, USA. [4] Laboratory for Atmospheric and Space Physics, University of Colorado, Boulder 80309 CO, USA. [5] NASA Langley Research Center, Hampton 23681 VA, USA. [6] Department of Atmospheric Science, University of Wyoming, Laramie 82071 WY, USA. [7] Present address: Department of Atmospheric and Oceanic Sciences, University of Colorado, Boulder 80309 CO, USA. Correspondence and requests for materials should be addressed to J.H.J. (email: Jonathan.H.Jiang@jpl.nasa.gov)

Atmospheric aerosols are considered to have the potential to inhibit or invigorate convective cloud development[1]. For absorbing aerosols, the traditional thinking is they may inhibit convection by blocking the radiation reaching the surface, enhancing the low-level stability. For example, the effect of Asian brown cloud upon rainfall over India for 1940–2040 has been simulated[2] by a coupled ocean-atmosphere model, from which absorptive aerosols warm the first several kilometers of the temperature profile, thereby stabilizing the lower troposphere. This stabilization produces an inhibition of convection, and even a weakening of summer monsoon systems in Asia[3,4]. It is conjectured that droughts in South Asia may be more likely to occur in the coming decades of the 21st century[5,6]. In addition, the regional circulation could be modified by absorbing aerosol[7]. It was found that the Hadley circulation can be weakened or expanded by absorbing aerosol, in contrast to the effects of other forcing agents such as greenhouse gases and sulfate aerosol[7,8]. It has also been hypothesized that, under a highly polluted condition, the aerosol radiative effect reduces the convective available potential energy (CAPE) and suppresses deep convection, overriding the aerosol microphysical invigoration effect, and leading to a net weakening effect on the cloud development[9,10].

On the other hand, an increase in aerosol concentrations and thus cloud condensation nuclei (CCN) may suppress warm rain processes, permitting more liquid water to reach the freezing level, and thereby enhance latent heat release in the upper portion of clouds, known as "cloud invigoration"[10–12]. For absorbing aerosols, the traditional suppression effect is also challenged by more complicated mechanisms. It is suggested that the aerosol heating near the top of the planetary boundary layer (PBL) can stabilize the PBL, increasing convection inhibition (CIN) within the PBL but enhancing CAPE above the PBL[13]. Over a longer timescale, the suppression of the shallow convection due to absorbing aerosols can postpone the release of energy and moisture, thus feeding and enhancing deep convection later[14,15].

Either cloud inhibition or invigoration could result in changes in the altitude of a convective cloud. The impact of such cloud height changes on the radiative budget of the atmosphere can be illustrated by considering the changes in the blackbody emission of optically thick clouds whose cloud tops are near 13 km altitude. With a temperature lapse rate of $-7\,\mathrm{K\,km^{-1}}$ near 13 km, the blackbody $\sigma T^4$ emission (with $\sigma$ the Stefan–Boltzmann constant) associated with cloud tops near 14 km are $13.8\,\mathrm{W\,m^{-2}}$ lower than that at 13 km. Therefore, quantification of how aerosols affect the development of convective clouds is extremely important to our understanding of the aerosol effects on weather and climate.

However, it is well recognized that quantification of how aerosol impacts clouds is not an easy task[16]. Difficulties often arise due to the potential dynamical feedbacks in a mesoscale cloud system, which could dampen aerosol invigoration or suppression effect on individual clouds; or due to the lack of observations of cloud life cycles owing to the snapshot nature of available satellite observations[17]. A major challenge is to isolate the effects of different types of aerosol on different types of clouds.

Satellite measurements from CALIPSO and CloudSat provide a closely collocated aerosol and cloud dataset with both aerosol-type and cloud-type information. The CALIPSO aerosol profile data include seven aerosol-type classifications[18]: elevated smoke, polluted continental/smoke, polluted dust, dust, clean continental, clean marine and dusty marine. For brevity, we refer to the 'elevated smoke' type as 'smoke' and the 'polluted continental/smoke' type as 'polluted continental' in this study. The CALIPSO–CloudSat combined cloud profile data include eight cloud-type classifications[19]: Deep Convective, Cirrus, Nimbostratus, Altostratus, Altocumulus, Cumulus, Stratus and Stratocumulus. Figure 1 illustrates an example of co-located CloudSat cloud and CALIPSO aerosol measurements. More details are given in the Methods section. A few studies have analyzed aerosol–cloud interactions as a function of cloud types. For example, Christensen et al.[20] used the CloudSat data to quantify the aerosol indirect effects on deep convective clouds. However, this study did not consider different aerosol types.

In this article, we focus on deep convective clouds and analyze the CloudSat/CALIPSO datasets to quantify the different convective cloud heights developed in different aerosol environments, such as smoke, dust and polluted continental aerosols. We find that different types of aerosols exert different effects on the deep convective cloud heights and non-linearity prevails in the aerosol effects.

## Results

**Type-dependent aerosol effects.** The first question arises as how to accurately measure the changes in cloud height due to aerosol effects. Observed cloud top heights and cloud depths vary substantially due to the diverse environmental conditions they develop in. The high vertical resolution of CloudSat measurements allow us to calculate a mass-weighted altitude centroid that integrates over a range of altitude for a deep convective cloud. Since the cloud inhibition (invigoration) process implies a decrease (increase) in imparted energy release in the cloud development process, we can use the cloud ice water content (IWC, $\mathrm{mg\,m^{-3}}$) and altitude ($z$, m) weighted centroid to measure

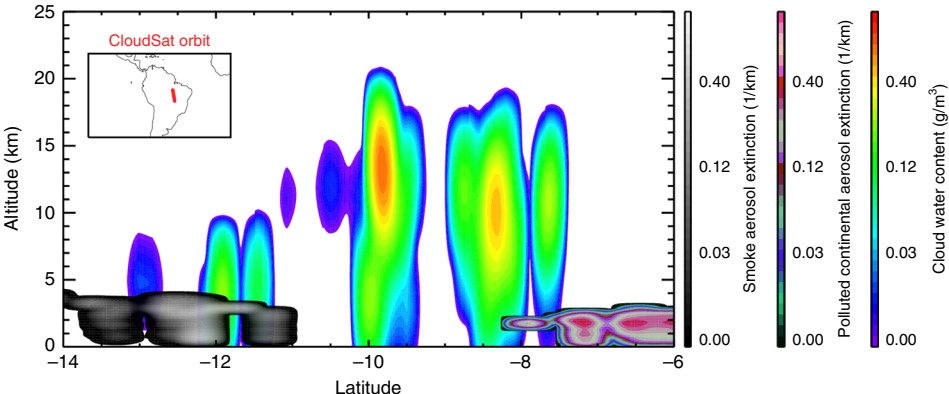

**Fig. 1** Curtain plot of clouds in different aerosol environments. Curtain plot of CloudSat/CALIPSO cloud water content and collocated CALIPSO smoke and polluted continental aerosol extinction profiles along an orbit over South America on 25 October 2007. The image is smoothed by a 12 km running window along the satellite tracks

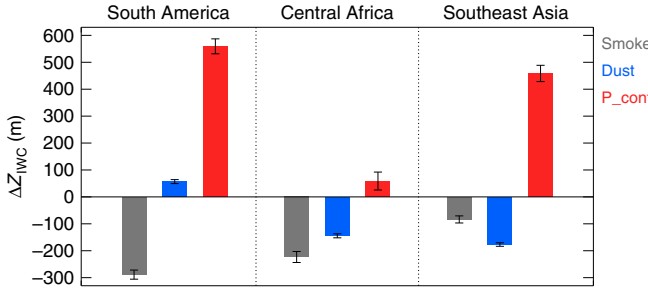

**Fig. 2** Changes of cloud altitude in different aerosol environment. Annual average changes of the altitude centroid for deep convective clouds, computed as $Z_{IWC}$ differences between clean environment and different aerosol environments: smoke aerosol (gray), dust aerosol (blue) and polluted continental aerosol (red) over the three selected regions: (left) South America, (middle) Central Africa and (right) Southeast Asia. The error bars denote the standard errors of the $\Delta Z_{IWC}$

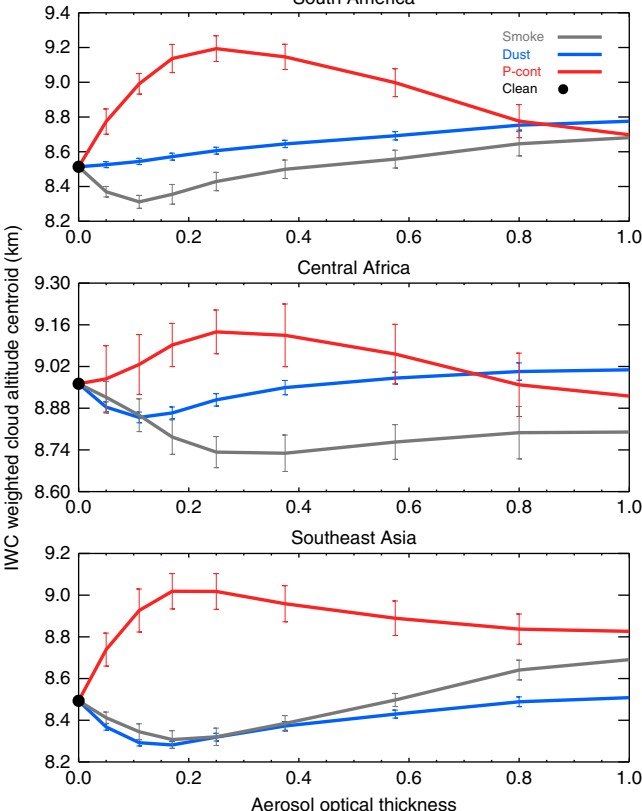

**Fig. 3** Non-monotonic responses of convective cloud to aerosol perturbation in different aerosol environments. IWC weighted altitude centroid for smoke aerosol (gray line), dust aerosol (blue line) and polluted continental aerosol (red line) as a function of aerosol optical thickness (AOT) for SAM (top panel), CAF (middle panel) and SEA (lower panel). The AOT includes aerosols above 500 m over the surface, see Methods section for details. The error bars denote the standard errors of the bin average

the changes in overall cloud ice mass being lofted into the air, and thus the strength of deep convection.

The altitude centroid[21] for each IWC profile, $Z_{IWC}$ is calculated from the expression:

$$Z_{IWC} = \int IWC \cdot z \cdot dz / \int IWC \cdot dz, \quad (1)$$

where $z$ is the altitude. If cloud inhibition (invigoration) is present, the $Z_{IWC}$ values decrease (increase) in an aerosol environment relative to a clean condition.

The IWC profile can be influenced by many factors, i.e., IWC ($z$, aerosol type, cloud type, region, season, meteorology). For this study, we focus our analysis on three aerosol types: smoke, dust and polluted continental aerosol; one cloud type: deep convective cloud; and three regions: South America (SAM, 0–30° S, 35° −80° W), Central Africa (CAF, 20° S–15° N, 10° W–50° E) and Southeast Asia (SEA, 0–35° N, 75°–125° E). These three regions are chosen considering the fact that the aforementioned three aerosols types are dominant (See Supplementary Figure 1 and the associated discussions), and convective activities are also abundant there.

Figure 2 shows the annual averaged $Z_{IWC}$ changes for convective clouds in different aerosol (smoke, dust, polluted continental) environments relative to clean conditions in the three regions. It is obvious that in smoke environments, the average convective cloud $Z_{IWC}$s are lower than those in clean conditions in all three regions. In polluted continental environments, the $Z_{IWC}$s are all higher than those in clean conditions in all three regions. When dust is the dominant aerosol type, the impact on the convective cloud ice altitude centroid shows a regional dependence. In SAM, dust contaminated $Z_{IWC}$s are higher than clean clouds. In CAF and SEA, $Z_{IWC}$s are lower in dust environments. In addition to the annual mean, the seasonal changes of $Z_{IWC}$ in different aerosol environments, different regions and different aerosol optical thickness (AOT) ranges are also studied (Supplementary Figures 2 and 3). The suppression effect of smoke and the invigoration effect of polluted continental on deep convection holds true in the seasonal mean, however, there are considerable variations with AOT and season due to the combinations of aerosol-type changes and meteorological variations, which will be discussed below.

**Non-monotonic cloud responses**. Figure 3 further shows, for each region, non-monotonic responses of convective cloud $Z_{IWC}$ to aerosol perturbations in three different aerosol environments, dominated by smoke, dust and polluted continental aerosols, respectively. For small aerosol loading up to AOT ~0.2–0.3, the convective cloud $Z_{IWC}$ decreases in the smoke aerosol environment, whereas conversely in the polluted continental environment $Z_{IWC}$ rapidly becomes higher with increasing AOT. This distinct difference suggests that smoke aerosol, which consists of mostly absorbing aerosols (black and organic carbon)[18,22,23], could act to stabilize the temperature profile and suppress convection as suggested by previous modeling studies and observations of Asian brown clouds[2]. In addition, smoke lifted by convection might also accelerate evaporation of cloud droplet or sublimation of ice crystals through absorptive heating[10], the so-called aerosol semi-direct effect. These combined effects could weaken the convective development and therefore lower $Z_{IWC}$ in the smoke environment. However, the behavior of convective clouds in the polluted continental environment (e.g., mixture of sulfate, nitrate and other pollutants), when AOT < 0.3, is consistent with the aerosol invigoration hypothesis[10] that an increase in CCN energizes deep convection through the latent heat release due to the freezing of a larger amount of water droplets, though it is possible that meteorological factors may also be of importance.

Under heavy aerosol loading conditions (AOT > ~ 0.3), the above effects are reversed: $Z_{IWC}$ increases with AOT in the smoke environment, whereas it decreases under polluted conditions, and both seem to stabilize at AOT > ~ 0.8. The monotonic increase of $Z_{IWC}$ with AOT in the thick smoke environment can be explained by the enhanced CAPE above the aerosol heating layer and

**Table 1 Total correlations between column AOT and IWC centroid, and the partial correlations with the effects of 12 meteorological parameters eliminated individually and simultaneously over all seasons**

| | South America | | | Central Africa | | | Southeast Asia | | |
|---|---|---|---|---|---|---|---|---|---|
| | Smoke | Dust | Polluted continental | Smoke | Dust | Polluted continental | Smoke | Dust | Polluted continental |
| Total correlation | **−0.076** | **0.030** | **0.14** | *−0.007* | *−0.020* | *0.010* | **−0.035** | **−0.13** | **0.071** |
| $RH_{850}$ | −0.072 | 0.028 | 0.15 | −0.008 | −0.018 | **0.030** | −0.030 | −0.12 | 0.067 |
| $RH_{500}$ | −0.072 | 0.028 | 0.15 | −0.008 | −0.018 | **0.030** | −0.030 | −0.12 | 0.067 |
| $RH_{350}$ | −0.070 | 0.038 | 0.17 | −0.023 | **−0.030** | 0.006 | −0.032 | −0.13 | 0.078 |
| LTS | −0.077 | 0.029 | 0.14 | −0.010 | −0.020 | 0.007 | −0.030 | −0.12 | 0.069 |
| $VV_{500}$ | −0.084 | 0.015 | 0.13 | −0.007 | −0.021 | 0.008 | −0.046 | −0.14 | 0.061 |
| $VV_{300}$ | −0.069 | *0.014* | 0.14 | −0.009 | −0.020 | 0.008 | −0.032 | −0.13 | 0.071 |
| $U_{300}$ | −0.099 | −0.046 | 0.11 | −0.007 | −0.020 | 0.009 | −0.076 | −0.18 | 0.070 |
| $U_{1000}$ | −0.068 | 0.031 | 0.14 | −0.010 | **−0.024** | 0.006 | −0.031 | −0.12 | 0.083 |
| $V_{300}$ | −0.091 | −0.004 | 0.13 | −0.005 | −0.021 | 0.010 | −0.054 | −0.14 | 0.078 |
| $V_{1000}$ | −0.077 | 0.031 | 0.14 | −0.005 | −0.020 | 0.010 | −0.027 | −0.13 | 0.079 |
| CAPE | −0.050 | −0.011 | 0.13 | *0.023* | *−0.001* | *0.021* | −0.080 | −0.17 | 0.067 |
| VWSH | −0.079 | 0.033 | 0.14 | −0.006 | **−0.025** | 0.008 | −0.036 | −0.13 | 0.071 |
| All parameters | **−0.033** | **−0.43** | **0.16** | *0.009* | *−0.016* | *0.015* | **−0.078** | **−0.18** | **0.064** |

AOT range is [0, 0.25]. Bold font indicates the significant agreement (same sign) between total and partial correlation, whereas non-bold font indicate the significant opposite signs between them. If total/partial correlation is not statistically significant at the 95% level, the corresponding font is italic

unconsumed instability and moisture from suppressed shallow convection[13–15]. On the other hand, the decrease of $Z_{IWC}$ with further increase of AOT in the heavy pollution environment (when the polluted continental aerosol dominates) suggests that substantial pollution could decrease the amount of sunlight reaching the surface, which in turn could weaken convection.

The non-monotonic response of cloud properties to aerosol perturbations is consistent with some earlier studies[24–26], in which there is often a turning point after which the aerosol effects reverse. This reflects the competing influences of aerosol microphysical and radiative effects. Our study further demonstrates the type dependency of this non-monotonic cloud response to aerosol forcing. The quantification of the aerosol–cloud relationships is difficult, due to the inadequacy of available tools to identify key responses to aerosol perturbations[16,27]. In addition, the feedbacks from environmental condition changes (e.g., moisture changes due to different evaporation efficiencies[28] or changes in mesoscale dynamics) could be non-monotonic as well[29].

**Influence of meteorological factors**. Changes in convective cloud vertical structure can also be caused by changes in large-scale meteorological conditions[30] in addition to the aerosol microphysical and radiative effects. For example, cloud formation is usually positively correlated with relative humidity (RH) and large-scale vertical velocity, and negatively correlated with wind shear[31–33]. For convective cloud, the strength of convection is also proportional to the buoyancy of an air parcel when it is lifted, which is measured by the CAPE[34].

To explore the meteorological impacts on the $Z_{IWC}$, we calculate the partial correlation between AOT and $Z_{IWC}$. The partial correlation is a measure of the linear dependence between two variables where the influence from possible controlling variables (meteorological parameters in this case) is removed[35–38]. More details of this method are given in the Supplementary section. In general, if the partial correlations are similar to the corresponding total correlation (at least they have the same sign), the correlation between AOT and $Z_{IWC}$ exists regardless of the effects of certain meteorological parameters. In other words, the meteorological covariations are not likely a main reason for the correlations between AOT and

$Z_{IWC}$. On the contrary, if the partial and total correlations have different signs, the effect of meteorological covariations is considered to dominate over the aerosol effects.

Twelve meteorological parameters are taken into account in this analysis. They include: $RH_{850}$, the RH at 800 hPa; $RH_{500}$, the RH at 500 hPa; $RH_{300}$, the RH at 350 hPa; LTS, the lower troposphere stability; $VV_{500}$, the vertical pressure velocity at 500 hPa; $VV_{300}$, the vertical pressure velocity at 300 hPa; $U_{300}$, the U-component of winds at 300 hPa; $U_{1000}$, the U-component of winds at 1000 hPa; $V_{300}$, the V-component of winds at 300 hPa; $V_{1000}$, the V-component of winds at 1000 hPa; CAPE; VWSH, the vertical wind shear at potential vorticity surface of $2 \times 10^{-6} \, km^2 \, kg^{-1} \, s^{-1}$. The results of regional-specific partial correlation coefficients with each or all of them removed are summarized in Table 1.

In South America and Southeast Asia, the annual total and partial correlations are generally similar for smoke and polluted continental aerosol, indicating that meteorological covariations are not likely a major reason for the correlations between AOT and $Z_{IWC}$. Also, the seasonal total and partial correlations (see Supplementary Table 1) are generally of the same sign as the annual results, except for a couple of cases where the correlations are very weak (e.g., smoke in the spring for South America). Therefore, the invigoration effect of polluted continental and the inhibition effect of smoke appear to be robust, although meteorological conditions may play a role in certain situations. In Central Africa, however, the impact of meteorological conditions on the annual mean is much larger and dominate the aerosol effects, as indicated by the different signs between the annual mean total correlations and partial correlations, as well as the presence of insignificant correlation coefficients. A closer look at the aerosol effects and meteorological influence in different seasons of Central Africa (See Supplementary Table 1) reveals the existence of the contrasting aerosol effects over different seasons. Both smoke and polluted continental aerosols show a significant suppressing effect on convection during winter, but their signs are reversed in the fall, resulting in a net insignificant aerosol effect in the annual mean. A comparison of Table 1 and Supplementary Table 1 suggests that the seasonality in meteorology might be key for Central Africa. The aerosol effects manifest on the seasonal timescale, but meteorology dominates on the annual mean.

The above analyses demonstrate the complexity of the AOT–$Z_{IWC}$ relationship in different seasons and meteorological conditions. In some places, meteorology affects convection in the same direction as aerosol, but in other places it might work in the opposite direction. However, the bulk behaviors of the inhibition effect of smoke and the invigoration effect of polluted continental aerosol shown in the annual mean (Fig. 2) and seasonal means (Supplementary Figure 3) are robust no matter what meteorological factors are considered. This is especially true for the smoke aerosol. The mechanisms for the observed relations should be identified through detailed cloud-resolving modeling studies in the future.

## Discussion

Impacts of different types of aerosols on deep convective clouds are examined in this study based on the latest available satellite data. Capitalizing on the aerosol speciation capability of CALIPSO and vertical profiling capability of CloudSat, we calculate changes in convective cloud altitudes weighted by ice water content ($Z_{IWC}$) as a function of AOT for smoke, dust and polluted continental aerosol types over South America, Central Africa and Southeast Asia. The $Z_{IWC}$ and AOT are calculated from the latest version of data from the CloudSat and CALIPSO experiments. We demonstrate that the impacts of different aerosol types on convective cloud development are substantially different. For the smoke aerosol environment, the $Z_{IWC}$ decreases with small aerosol loading up to AOT ~0.2, then gradually increases as aerosol loading increases. Conversely in the polluted continental environment, the $Z_{IWC}$ increases with mild aerosol loading, but decreases with further aerosol enhancement. The influence of dust on convective cloud ice has a strong regional dependence compared with the smoke and pollution impacts. Our findings provide observational evidence that the aerosol "inhibition" and "invigoration" processes, previously hypothesized in the literature, strongly depend on aerosol type and concentration. In a light smoke environment, aerosols suppress deep convection, producing ice clouds with lower altitude centroid. If smoke is substantial, shallow convection can be totally shut down, but subsequent deep convection can become even stronger due to the unconsumed CAPE[15,39,40], especially above the PBL[13]. The reverse is true for aerosols from anthropogenic pollution, in which lightly polluted air invigorates convection and produces ice clouds at higher altitude. In a heavily polluted environment, convection is weakened by decreasing the amount of sunlight reaching the surface. The aerosol effects on convective clouds over Central Africa are generally less pronounced than those over South America and Southeast Asia.

In South America, dust tends to invigorate convection on the annual mean (Fig. 1) but have mixed results when meteorological factors are considered (Table 1). In Southeast Asia and Central Africa, dust aerosols behave like smoke that suppress convection. We note that a substantial amount of South American dust originates in the Sahara[41]. In contrast, the Southeast Asia and Central Africa dusts originate from the same continent. Because dust near the source is likely to be at a lower altitude than dust transported over a long distance, the South American dust likely has a different altitude profile compared with the Southeast Asia and Central Africa dust. Dust at lower altitudes influences the temperature profile and atmospheric stability in a manner different from dust at higher altitudes. This likely explains the differences in the effect of dust on $Z_{IWC}$ in South America when compared with Southeast Asia and Central Africa.

Our study presents interesting results, which can be compared with previous studies. Koren et al.[24] used MODIS aerosol and cloud top pressure data to study how cloud top pressure changes

along with aerosol variations. Figure 2 of their study indicates that cloud top height first increases and then decreases as aerosol increases. This behavior was attributed to the enhanced aerosol that reduces the amount of surface illumination. The polluted continental curves in our Fig. 3 display similar characteristics. However, we find that clouds in smoke, polluted continental and certain dust environments behave differently, suggesting that cloud heights are also sensitive to the aerosol type. Figure 10 of Koren et al.[32] indicates that cloud top pressure changes, due to changes in aerosol, also are dependent upon RH and the convective regime (i.e., the $\omega$ pressure vertical velocity). They found that cloud heights are higher for larger RH values. Our study finds that the cloud heights are sensitive to aerosol type, regardless of moisture content in the atmosphere. Massie et al.[21] analyzed the vertical shapes of IWC profiles as a function of MODIS aerosol, ozone monitoring instrument absorbing aerosol and Microwave Limb Sounder CO at 215 hPa (a smoke proxy). They suggested that differences in the IWC vertical shape profiles for deep convective clouds are consistent with the assumption that absorptive aerosol inhibits convective cloud development. This aspect of their results is confirmed by our study's analysis. Previous studies[25,42] show aerosol can either suppress[24,43,44] or invigorate[12,45–47] the development of deep convective clouds. Our analyses highlight, in addition to meteorological factors, that the aerosol type is one of the key factors that determine the aerosol effects on convective clouds.

It is worth noting that caveats exist when using CloudSat and CALIPSO in the study. For example, polar-orbit satellite data, measured at fixed 1:30 a.m./p.m. observation times, only provide an instantaneous relationship between aerosol and clouds, and do not address aerosol effects on cloud lifecycle and time-dependent mesoscale convection systems. However, as current global climate models do not accurately depict deep convective cloud structure and lack a comprehensive representation of aerosol radiative, CCN and ice nuclei (IN) effects, especially on the convection-resolving scale[46], satellite observations do provide a unique opportunity to address the aerosol–cloud interaction challenges for different regions around the world. Our observational findings of aerosol-type impacts on convective clouds serve as valuable constraints on the modeling of aerosol–cloud interactions.

## Methods

**Data**. The newest data versions are used in this study. For CALIPSO, we analyze Version 4 Level 2 aerosol profile data (CAL-LID-L2-05 kmAPro). For CloudSat, we analyze version R04 combined Level 2 cloud profile data (2C-ICE and 2B-CLDCLASS-LIDAR). We primarily use aerosol optical thickness (i.e., we call it AOT here) and aerosol-type information from CAL-LID-L2-05 kmAPro data, IWC from 2C-ICE data, and cloud-type information from 2B-CLDCLASS-LIDAR data.

The features identified by CALIPSO are first classified as aerosol or cloud using a cloud–aerosol discrimination (CAD) algorithm[48]. The level of confidence in the aerosol–cloud classification is indicated by a CAD score, with negative values (−100 to 0) for aerosol and positive values (+100 to 0) for cloud. After an aerosol layer is identified, the scene classification algorithm further categorizes the aerosol layer to 1 of 7 aerosol types, by using input parameters including altitude, location, surface type, volume depolarization ratio and integrated attenuated backscatter measurements[18]. The new Version 4 Level 2 CALIPSO aerosol data products, released in November 2016, include substantial improvements to the aerosol subtyping and lidar ratio selection algorithms over the prior Version 3 product. The aerosol detection thresholds of this product are discussed and quantified in the literature[49] and are taken into account by our analyses.

The 2B-CLDCLASS-LIDAR product classifies cloud scenarios into eight cloud types by using vertical and horizontal cloud properties, the occurrence and intensity of precipitation, cloud temperature and cloud phase[50]. A cloud type is assigned to each cloud layer, so clouds in a single radar profile may be partitioned into several types if there are well separated cloud layers.

The CALIPSO aerosol profile data have a uniform spatial resolution of 60 m vertically and 5 km horizontally, over a nominal altitude range from the surface to 20 km. The footprint for a single profile of CloudSat observation is approximately 1.3 km across-track by 1.7 km along-track, with along-track sampling spaced every 1.1 km[51]. The cloud measurements are reported on an increment of ~240 m with 125 vertical layers. The time period for this study is from June 2006 to December

2010, since the CALIPSO/CloudSat combined cloud profile data are publicly available only for this period. For the meteorological fields, we use CloudSat auxiliary ECMWF data (ECMWF-AUX), which provide the vertical profiles of pressure, temperature and specific humidity from the surface to the upper troposphere co-located with each CloudSat profile measurement. All the meteorological parameters are calculated from ECMWF-AUX data for each CloudSat profile.

**Data collocation**. To co-locate daily CALIPSO aerosol with CloudSat cloud observations, we first identify a $1° × 1°$ grid box that is centered on each CloudSat profile, then find all the CALIPSO aerosol profiles within this grid box. The occurrence frequency of each aerosol type is calculated as the number of each aerosol-type samples divided by the total number of all aerosol-type samples for all the co-located aerosol profiles. To reduce the effects of surface contamination in CALIPSO aerosol data, all aerosol samples with an altitude <500 m are ignored. The co-located AOT is the average of column AOT for all the co-located aerosol profiles. If no aerosols (above 500 m) are detected within the $1° × 1°$ grid box, it is defined as a "clean" (i.e., no aerosols) environment case; if an aerosol type has occurrence frequency larger than 90%, then it is defined as an "aerosol" environment case dominated by that particular type. This type definition approach is also applied to CloudSat cloud types. Since this study focuses on deep convective clouds, we only select the CloudSat profiles where deep convective cloud is the dominant cloud type. CloudSat measurements within the lowest kilometer are affected by ground contamination[19,52], thus we limit our analysis to cloud profiles above 1 km. The IWC data from 2C-ICE are interpolated into 15 altitude bins with 1 km interval extending from 1 km to 16 km altitude.

Whenever a deep convective cloud profile is identified in one of the three aerosol types (smoke, dust and polluted continental aerosol) environment, we calculate the altitude centroid $Z_{IWC}$ for the IWC profile using Eq. (1). Then, we calculate the seasonal and annual average of $Z_{IWC}$ for each target region using all the available data. The impacts of different aerosol types on $Z_{IWC}$ is determined by the difference of $Z_{IWC}$ between each aerosol-type environment and the clean environment (i.e., case without any aerosol).

**Partial correlation between AOT and $Z_{IWC}$**. To exclude the impact of meteorological covariation, we calculate the partial correlation between AOT and $Z_{IWC}$. Let $\mathbf{X}$ denote a vector of meteorological parameters, the effects of which we would like to eliminate. The partial correlation between AOT and $Z_{IWC}$, eliminating the effects of $\mathbf{X}$, is:

$$\rho_{AOT-Z_{IWC}·\mathbf{X}} = \frac{\sigma_{AOT-Z_{IWC}·\mathbf{X}}}{\sigma_{AOT·\mathbf{X}}\sigma_{Z_{IWC}·\mathbf{X}}} \qquad (2)$$

where $\sigma_{AOT-Z_{IWC}·\mathbf{X}}$ is the conditional covariance between AOT and $Z_{IWC}$, eliminating the effects of $\mathbf{X}$; $\sigma_{AOT·\mathbf{X}}$ is the square root of the conditional variance of AOT, eliminating the effects of $\mathbf{X}$; $\sigma_{Z_{IWC}·\mathbf{X}}$ is the square root of the conditional variance of $Z_{IWC}$, eliminating the effects of $\mathbf{X}$. More details of the calculation method for partial correlation are described in other references[35–38].

The total and partial correlations are calculated using the samples with $0 ≤ AOT ≤ 0.25$ and averaged for every 500 samples. Note that the correlation coefficients among aerosol, meteorological factors and $Z_{IWC}$ are generally low in both Table 1 and Supplementary Table 1, due to noise in the transient observations and spatial scale differences between satellite (~ 1 km pixel level) and reanalysis (~1° gridded). For example, the total correlation is only 0.092 between $RH_{350}$ and $Z_{IWC}$ and 0.28 between CAPE and $Z_{IWC}$.

## Data availability

The data used for this study can be downloaded from the NASA Distributed Active Archive Centers at https://earthdata.nasa.gov/about/daacs.

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

## Acknowledgements

This research is supported by NASA CCST and ACMAP programs. The work was conducted at the NASA sponsored Jet Propulsion Laboratory, California Institute of Technology, under contract with NASA. We also acknowledge the support by the NASA Langley Research Center, University of California, Los Angeles, University of Colorado, Boulder and the University of Wyoming.

## Author contributions

J.H.J. designed the analysis and wrote the paper. J.H.J. and L.H. processed, computed and analyzed the CloudSat and CALIPSO data. H.S., Y.W., S.M. and B.Z. contributed to the data analysis and writing of the paper. A.O. and Z.W. provided guideline for CloudSat and CALIPSO data processing and usage. Everyone edited the manuscript.

## Additional information

**Competing interests:** The authors declare no competing interests.

