## [Peer Review File · Nature Communications]

Reviewer #1 (Remarks to the Author):

This manuscript presents a new attempt to disentangle the various competing aerosol effects on convection based on satellite observations, using active sensing from the CALIPSO and CloudSat platforms to discriminate between aerosol types and to probe cloud phase and vertical structure. Aerosol–convection interactions are some of the most uncertain aspects of the climate impact of aerosol, so any improvement in understanding may have significant implications.

The analysis focusses on the correlation of column-integrated aerosol optical thickness (AOT) with the vertical centroid of ice mass (Z_{IWC}). Different relationships are shown for different aerosol types, with most cases showing nonlinear (and sometimes non-monotonic) responses to increasing AOT. The robustness of the results, however, appears to be significantly overstated, with many of the aerosol types showing quite different turning-point behaviour in the AOT– Z_{IWC} response when stratified by RH or CAPE, and different responses in different seasons. This suggests that the authors claim to "untangle the mystery of opposite signs of aerosol effects" and provide "unambiguous references for modeling aerosol-cloud interactions" is not fully realised.

The approach also does not directly address the question of causality: although it is implied that the variation of Z_{IWC} with AOT is the result of aerosol-induced convective invigoration or suppression, there is little or no discussion of how confounding factors can be excluded (e.g. seasonal movement of the ITCZ bringing different convective regimes and aerosol conditions in tandem, or causality in the opposite direction where convective regimes control winds or fire risk and thus dust/smoke emissions rather than the other way around).

The manuscript is well written and has the potential to be a significant contribution with further development, but given the above issues I would recommend that it is rejected at this point and the authors encouraged to resubmit if the inconsistencies in the stratified responses can be explained, and the questions of causality and confounding factors adequately addressed.

A number of more specific comments are listed below:

p.1, line 11: "convective clouds produce most of the global precipitation" is perhaps too strong, unless quantified with references; I would suggest "a significant proportion of" instead.

p.1, lines 27–31: as discussed above, these conclusions are much more confident than supported by the evidence presented.

p.2, lines 2–3: "aerosols are considered to have the ability to inhibit or invigorate..." suggests these effects are both well demonstrated to occur in nature. There are many potential such effects, but they remain mostly conjectures requiring further evidence that they occur outside of idealised studies.

p.2, lines 5–16: are these model responses robust between different models?

p.2, lines 16: "overriding the aerosol microphysical invigoration effect" suggests that such an effect is definitely present, but no argument to support this is referenced and (unlike inhibition) the potential mechanism hasn't been described at this point in the text.

p.3, lines 3–5: it is also worth addressing:

(a) the difficulties which arise due to the potential for cloud-system, mesoscale and larger feedbacks to dampen the impact of any aerosol invigoration or suppression effect on individual clouds (e.g. Stevens and Feingold 2009 is cited as [11] which has this as a central theme but this is not discussed); and

(b) those which arise in remote-sensing studies because it's unclear whether the snapshot of observed aerosol represents the environment in which the convective cloud forms or that left behind after the convection has lofted or removed a significant amount of aerosol (see e.g. Gryspeerd et al., 2015, doi:10.5194/acp-15-7557-2015).

p.4, line 1: presumably part of the reason for choosing these regions is also their significance in terms of global convective activity?

p.5, line 13: in addition to aerosol lifted by convection, the prior altitude of the aerosol layer in the environment can play an important role especially for absorbing aerosol. Given the study is based on CALIOP aerosol data, which has good vertical resolution, it is a shame that the analysis doesn't explicitly consider the different responses due to aerosol at different levels.

p.6, lines 2–3: "is consistent with the aerosol invigoration effect" is true, but it is important to acknowledge that this in no way demonstrates that this effect is in fact the mechanism.

p.6, lines 10–11: the suggestion that the pollution is dimming enough to weaken the convection is highly speculative without evidence to demonstrate the mechanism.

p.6, line 18: the environmental changes are not restricted to moisture, but also mesoscale dynamics etc.

p.6, lines 20–21: changes due to large-scale conditions may also be the result of large-scale aerosol–radiation and aerosol–cloud interactions.

p.7, lines 8–9: the dominance of increased CAPE doesn't in itself explain the reversed smoke effect – it would only explain a reduced or absent effect.

p.7, lines 27, 30: again, it is not demonstrated that these mechanisms (unconsumed CAPE, decreased sunlight) are in fact the cause of the effects seen.

p.7, lines 31–32: can it be shown whether the different dust behaviours are due to differences in the dust itself, the convective regimes in question, or other external factors?

p.8, lines 10–11: "more sensitive to aerosol type, not just RH" suggests that aerosol type has a larger impact on convection than RH, which seems unlikely and is probably not the intended meaning. Please clarify.

p.8, line 19: different-signed effects are also still seen for the same aerosol type in some cases between regions, seasons and RH/CAPE bins, so these are not fully disentangled.

p.8, line 22: it is not just the individual cloud lifecycle, but also that of cloud systems and the mesoscale that may be relevant here.

p.9, lines 25–28: what kind of data is ECMWF–AUX? Is this pure model, pure observations, operational analysis, reanalysis (e.g. ERA-Interim) or something else?

p.10, line 2: the exclusion of aerosol below 500m risks excluding **exactly** the near-surface boundary-layer aerosol which is being entrained into the convective cloud base and thus driving any effects based on CCN activity. This is likely to be especially true in the context of convection out of a shallow but heavily-polluted boundary layer. The potential impacts on the results should be discussed.

p.10, lines 22–24: this mentions computer code used which is available from the authors; however the policy checklist states that no custom code is involved. This should be clarified and any additional policy documentation completed for the code if necessary.

p.13, lines 24–25 and 32–33: these seasonal variations in dust may be correlated with seasonal variations in convection for reasons unrelated to aerosol–convection interactions (e.g. large-scale circulation patterns).

p.15, lines 9–10: the text notes the possible combination of aerosol and meteorological changes, but the potential impacts of such non-causal covariations on the results needs to be considered in much more detail as discussed above.

pp.16–17, Figs S4–S7: the qualitative shapes of these curves in most cases do **not** look consistent between low/high RH or CAPE cases. A clearer explanation of the particular features which **are** consistent amongst the many differences is needed.

Reviewer #2 (Remarks to the Author):

This is an appealing paper that pulls together and seems to make sense of several previous studies of the effect of changes in aerosol on deep convection. In many ways it's a fairly straightforward paper that essentially presents a slightly more in-depth analysis of satellite data used by other studies, but makes the critical step of accounting for the aerosol type. It's mostly a satellite analysis with quite a lot of speculation about what might cause the different responses to different aerosol types and loadings. The speculations make sense (and are broadly supported by other studies), but the theoretical understanding and interpretation of the results is a bit unsatisfying.

My main concern is that the meteorological environment is not very thoroughly explored as a potential compounding variable. The analysis is performed for two RH and CAPE ranges. However, I'm not completely convinced that the authors properly account for the possibility that aerosol loading can correlate with CAPE, CIN, RH, etc. To be convincing I suggest the authors show a scatterplot of aerosols versus the meteorological factors that could influence the IWC centroid. This is not the same thing as showing the same centroid v aerosol relationship at different RH. What does aerosol v RH look like when $d\text{centroid}/d\text{aerosol}$ is positive and negative? This should be repeated for other factors too. To be honest I would be surprised if these relationships reversed at a particular aerosol load (sufficient to provide an alternative explanation of the data), but the study would be considerably strengthened if you carried out this additional analysis.

The discussion and comparison with previous studies on p8 is quite unsatisfying. I was left really not understanding why your study differed. In particular, the following statement needs to be properly supported: "In previous studies¹⁸, there are many contradictory findings (of opposite sign) in regard to aerosol-cloud effects. Our analyses help to disentangle this problem, since different signed aerosol-cloud effects are associated with different aerosol types." I'd like to see more than "helps to disentangle". You should point out exactly how each study reached an apparently contradictory result that can now be understood in the light of different aerosol types. There are not that many. Perhaps a table in the SI would be a good way of showing this case by case information.

There are quite a few typos and poor uses of English that I guess can easily be taken care of.

There are a couple of relevant papers that could be cited if space allows:

Wall et al. (2014) highlighted the importance of environment in controlling the response to aerosols: <https://doi.org/10.1175/JAS-D-13-0158.1>

Cui et al. (2011) showed the non-monotonic effect of increases in aerosols for a range of cloud types, CAPE etc: <https://doi.org/10.5194/acp-11-3495-2011>

Reviewer #3 (Remarks to the Author):

Do Aerosols Inhibit or Invigorate Convection? A Tale of Three Aerosol Types

By, Jonathan H. Jiang¹, Hui Su¹, Lei Huang², Yuan Wang³, Steven Massie⁴, Bin Zhao², Ali Omar⁵, Zhien Wang⁶

General Comment

The analysis presented here provides some new lines of evidence for the complex aerosol responses in convective clouds observed in many previous studies. Specifically, this study reveals how the sign of the convective invigoration response can be influenced by aerosol type. The authors draw on well-established methodology and state-of-the-art satellite datasets. While the CALIOP retrieval of aerosol type may be imperfect (and further discussion of this is warranted) the analysis nicely demonstrates that these opposing responses are robust through the in-depth analysis of meteorological regimes (CAPE and relative humidity) provided in the supplementary section. Overall, the paper is well written and the presented arguments are supported by the figures and conclusions.

Other comments

How reliable is the CALIOP aerosol type detection? The results of this study largely hinge on the lidar's ability to classify aerosol correctly. I therefore would have expected more discussion on this point. Several studies examine the aerosol type retrieval from CALIOP using a variety of sources of validation. For example, the Cape Verde region aerosol type is reliably classified (Tesche et al. 2013). However, Wu et al. (2014) results show that the detection of very dense smoke layers are sometimes classified as cloud and thin aerosol layers are not commonly detected. Kacenelenbogen et al. (2014), JGR compare CALIOP with the High Spectral Resolution Lidar (HSRL) flown on the NASA Langley aircraft in coincident locations as CALIOP. Kacenelenbogen et al. (2014) demonstrates that the CALIOP retrieved aerosol type is actually fairly different from the one derived from HSRL and CALIOP tends to mostly misclassify smoke, polluted continental, dust, and clean marine aerosols. Mielonen et al. (2009) shows that CALIOP agrees with AERONET aerosol type about 70% of the time. Aerosol type from CALIOP has also been compared to AeroCom models showing weak sensitivity to the extinction mean height diagnostic in industry and maritime locations but high sensitivity in African and Chinese dust regions (Koffi et al. 2012). From this literature review it is clearly difficult to determine exactly the accuracy of the CALIOP aerosol type retrieval from these studies. Broadly speaking, this is a challenging issue. Nonetheless, I would recommend including a few sentences on the validation of CALIOP aerosol type so the reader understands more of the sources of uncertainty. I am also sure whether there may be potential issues with the aerosol typing beneath relatively thick cirrus or cloud layers but this should also be investigated to ensure the retrievals are suitable in these locations too.

Pg1 L21: “regional” to “regionally”

Pg2 L21: May want to mention that ultrafine aerosol particles also have recently been shown to enhance convection (Fan et al. 2018).

Pg3 – L4-5 and L9: Other studies have also sorted responses by cloud types. For example, Christensen et al. (2016) used the same CloudSat (CLDCLASS-LIDAR) data to quantify aerosol indirect radiative effects for deep convective clouds. Similarly, cloud types retrieved from MODIS were used in Oreopoulos et al. (2017). These studies also found non-linear aerosol responses of the deep convective clouds (for example as you discuss on pg 17 L8) but did not sort by aerosol type as was uniquely conducted in your study.

Pg3 L24-27: I would like to point out another study which uses a similar method to quantify the shift in the altitude centroid of deep convective clouds. Storer et al. (2014) used CloudSat to demonstrate that polluted clouds have a higher reflectivity centroid (center of mass concept) compared to unpolluted clouds over the central Atlantic Ocean.

Pg4 – L10: A pesky little point but I would replace the word “clear” with “obvious” or “evident” to avoid implying that “smoke” is actually “clear.”

Figure 1: What is considered polluted versus unpolluted? The following paragraph suggests $AOT < 0.2$ is unpolluted but the supplementary materials suggest it is the case when “no aerosols (above 500m)” are detected {Pg10 L3-4}.

Pg6 L10: This sentence should probably specify which type, smoke or polluted continental aerosols, is being considered here. The previous sentence refers to heavy smoke aerosols. I assume this sentence is referring to polluted continental aerosols although it is not specifically stated.

Pg10 L3-4: Can the definition of “clean” be clarified? Is it possible that “no aerosols” will be detected above 500 m if the CALIOP lidar is fully attenuated by an overlying cirrus cloud? In such a case, I could imagine the air being very “polluted” but these aerosols would be missed and classified as “clean” in the analysis simply because the lidar has no sensitivity due to attenuation. How are these cases treated?

Figure S2: It would be noteworthy to point out that Fig S2c is the same as Figure 1, just shown on different scales.

References

Christensen, M. W., Y.-C. Chen, and G. L. Stephens (2016), Aerosol indirect effect dictated by liquid clouds, *J. Geophys. Res. Atmos.*, 121, doi:10.1002/2016JD025245.

Fan J, Rosenfeld D, Zhang Y, Giangrande S E, Li Z, Machado L A, Martin S T, Yang Y, Wang J and Artaxo P 2018 Substantial convection and precipitation enhancements by ultrafine aerosol particles *Science* 359 411–418

Kacenelenbogen, M., J. Redemann, M. A. Vaughan, A. H. Omar, P. B. Russell, S. Burton, R. R. Rogers, R. A. Ferrare, and C. A. Hostetler (2014), An evaluation of CALIOP/CALIPSO's aerosol-above-cloud detection and retrieval capability over North America, *J. Geophys. Res. Atmos.*, 119, 230–244, doi:10.1002/2013JD020178.

Koffi, B., et al. (2012), Application of CALIOP layer product to evaluate the vertical distribution of aerosols estimated by global models: AeroCom phase I results, *J. Geophys. Res.*, 117, D10201, 2012 doi:10.1029/2011JD016858.

Mielonen, T., A. Arola, M. Komppula, J. Kukkonen, J. Koskinen, G. de Leeuw, and K. E. J. Lehtinen (2009), Comparison of CALIOP level 2 aerosol subtypes to aerosol types derived from AERONET inversion data, *Geophys. Res. Lett.*, 36, L18804, doi:10.1029/2009GL039609.

Oreopoulos L, Cho N, Lee D, Kato S (2016) Radiative effects of global MODIS cloud regimes. *J Geophys Res Atmos* 121. doi: 10.1002/2015JD024502

Storer, R. L., S. C. van den Heever, and T. S. L'Ecuyer (2014), Observations of aerosol-induced convective invigoration in the tropical east Atlantic, *J. Geophys. Res. Atmos.*, 119, 3963–3975, doi:10.1002/2013JD020272.

Tesche, M., U. Wandinger, A. Ansmann, D. Althausen, D. Müller, and A. H. Omar (2013), Ground-based validation of CALIPSO observations of dust and smoke in the Cape Verde region, *J. Geophys. Res. Atmos.*, 118, 2889–2902, doi:10.1002/jgrd.50248.

Wu, Y., Cordero, L., Gross, B., Moshary, F., and Ahmed, S.: Assessment of CALIPSO attenuated backscatter and aerosol retrievals with a combined ground-based multi-wavelength lidar and sun-photometer measurement, *Atmos. Environ.*, 84, 44–53, 2014.

Reviewers' comments:

Reviewer #1 (Remarks to the Author):

This manuscript presents a new attempt to disentangle the various competing aerosol effects on convection based on satellite observations, using active sensing from the CALIPSO and CloudSat platforms to discriminate between aerosol types and to probe cloud phase and vertical structure. Aerosol–convection interactions are some of the most uncertain aspects of the climate impact of aerosol, so any improvement in understanding may have significant implications.

The analysis focusses on the correlation of column-integrated aerosol optical thickness (AOT) with the vertical centroid of ice mass (Z_{IWC}). Different relationships are shown for different aerosol types, with most cases showing nonlinear (and sometimes non-monotonic) responses to increasing AOT. The robustness of the results, however, appears to be significantly overstated, with many of the aerosol types showing quite different turning-point behaviour in the AOT– Z_{IWC} response when stratified by RH or CAPE, and different responses in different seasons. This suggests that the authors claim to "untangle the mystery of opposite signs of aerosol effects" and provide "unambiguous references for modeling aerosol-cloud interactions" is not fully realized.

Thank you very much for your helpful comments. We agree that the different turning-point of the AOT- Z_{IWC} relationship at different seasons and meteorological environments reflect the complexity of the nonlinear response to aerosol forcing. This reflects the competing influences of aerosol microphysical and radiative effects. Such non-linear cloud response to aerosol forcing is consistent with some earlier studies, but our study further demonstrates the non-linearity depends on aerosol type. Text is modified accordingly.

We also have conducted additional analysis in this revision to explore the meteorological factors that could influence the IWC centroid (see partial correlation analysis on page 7-8 and Supplemental Table S1). The results demonstrate again that the invigoration effect of polluted continental and the inhibition effect of smoke are robust, although meteorological conditions may play a role in certain situations.

We have revised the abstract, removing the statement that we have “untangled” the opposite sign mystery of the aerosol-cloud interaction. Instead, we state that “Our observational findings indicate that different aerosol types are one of the key factors in determining the aerosol effects on convective clouds.”

The approach also does not directly address the question of causality: although it is implied that the variation of Z_{IWC} with AOT is the result of aerosol-induced convective invigoration or suppression, there is little or no discussion of how confounding factors can be excluded (e.g. seasonal movement of the ITCZ bringing different convective regimes and aerosol conditions in tandem, or causality in the opposite direction where convective regimes control winds or fire risk and thus dust/smoke emissions rather than the other way around).

As an observation-based study, the only feasible way to prove causality is trying to exclude the possible meteorological factors other than aerosols. The variations due to seasonal movement of the ITCZ and thus different convective regimes have been taken into account when the AOT- Z_{IWC} relationship is analyzed for each season separately (Figure S3). The inhibitive effect of smoke is present during all seasons for all three geographical regions. We found aerosol type dependent

AOT- Z_{IWC} relations in almost all the regions and seasons, and the statistics is robust given the large number of data points subdivided in each data bin. Thus we are confident that the type-dependent aerosol AOT impact on convective cloud Z_{IWC} is significant. The convection induced fire emissions and wind-blown dusts are more associated with dry convection, rather than moist convection discussed here. The deep convective cloud (DCC) influence on aerosols can be excluded in our observed relationships, because DCCs typically produce heavy precipitation which tends to scavenge the aerosols, resulting in the same AOT- Z_{IWC} relationship between smoke and polluted continental aerosols.

The manuscript is well written and has the potential to be a significant contribution with further development, but given the above issues I would recommend that it is rejected at this point and the authors encouraged to resubmit if the inconsistencies in the stratified responses can be explained, and the questions of causality and confounding factors adequately addressed.

The questions of causality is complicated by difficult questions, the 1st on seasonal differences and the 2nd on meteorological covariation, which are actually closely related with each other. The differences in IWC centroid responses according to season can be partly explained by the different responses as a function of meteorological parameters. For example, in Central Africa, smoke suppresses convection in all seasons except summer. This is consistent with the fact that smoke suppresses convection when 850hPa RH < 80% but does not when 850 hPa RH > 80%. This helps to address the 1st question.

The 2nd question is to address why the responses of IWC centroid to AOT under different meteorological ranges are often of opposite sign in one region (especially Central Africa) but not in other regions. In some cases, especially in Central Africa, the meteorological effects seem very strong and exceeding the aerosol effects, while in South America and Southeast Asia the aerosol effects are unambiguous. Therefore, we have to accept that the strong effects of meteorological conditions may overwhelm the aerosol effects in some cases, especially in Central Africa. In these cases, the aerosol effects may not be cleanly isolated by our data analyses. This is also a reason for some of the contradicting results reported in previous studies.

We agree that all inconsistencies are likely not explainable without further theoretical study, such as detailed microphysics modeling. This is outside of the scope of the revised paper. We have, however, added additional analysis and added Table 1 on page 7-9 of the paper that addresses the confounding meteorological factors.

The observational findings of the type-dependent aerosol effects on convective clouds, presented in the revised paper, will generate sufficient interest in the modeling of the AOT- Z_{IWC} relationship.

A number of more specific comments are listed below:

p.1, line 11: "convective clouds produce most of the global precipitation" is perhaps too strong, unless quantified with references; I would suggest "a significant proportion of" instead.

Revised to "a significant proportion of" as suggested.

p.1, lines 27–31: as discussed above, these conclusions are much more confident than supported by the evidence presented.

We have revised the Abstract to read: "*Our observational findings indicate that different aerosol types are one of the key factors in determining the aerosol effects on convective clouds*". This statement is supported by the evidence presented in the revised paper.

p.2, lines 2–3: "aerosols are considered to have the ability to inhibit or invigorate..." suggests these effects are both well demonstrated to occur in nature. There are many potential such effects, but they remain mostly conjectures requiring further evidence that they occur outside of idealized studies.

We changed wording to "*aerosols are considered to have the **potential** to inhibit or invigorate...*"

p.2, lines 5–16: are these model responses robust between different models?

Yes, those aerosol-induced responses are reported by different models. Now we have added more references using different models to justify this point.

Added References:

Meehl, G., J. Arblaster, and W. Collins, 2008: Effects of black carbon aerosols on the Indian monsoon. *J. Climate*, 21, 2869–2882.

Allen, R. J., Sherwood, S. C., Norris, J. R., and Zender, C.S.: Recent Northern Hemisphere tropical expansion primarily driven by black carbon and tropospheric ozone, *Nature*, 485, doi:10.1038/nature11097, 350–353, 2012.

p.2, lines 16: "overriding the aerosol microphysical invigoration effect" suggests that such an effect is definitely present, but no argument to support this is referenced and (unlike inhibition) the potential mechanism hasn't been described at this point in the text.

Recent cloud-resolving simulations of both aerosol microphysical and radiative effects on a convective system over the US SGP [Wang et al., 2018] has demonstrated a different sign of the radiative effect for light absorbing aerosols in comparison to the CCN effect. This paper is referenced.

Added Reference:

Wang, Y., and Coauthors, 2018: Aerosol microphysical and radiative effects on continental cloud ensembles. *Adv. Atmos. Sci.*, 35(2), 234–247

p.3, lines 3–5: it is also worth addressing:

(a) the difficulties which arise due to the potential for cloud-system, mesoscale and larger feedbacks to dampen the impact of any aerosol invigoration or suppression effect on individual clouds (e.g. Stevens and Feingold 2009 is cited as [11] which has this as a central theme but this is not discussed); and (b) those which arise in remote-sensing studies because it's unclear whether the snapshot of observed aerosol represents the environment in which the convective cloud forms or that left behind after the convection has lofted or removed a significant amount of aerosol (see e.g. Gryspeerd et al., 2015, doi:10.5194/acp-15-7557-2015).

Thank you for the helpful comments. We added a few sentences to address these points: "*Difficulties often arise due to the potential dynamical feedbacks in a mesoscale cloud-system, which could dampen aerosol invigoration or suppression effect on individual clouds; or due to the lack of observations of cloud life cycles owing to the snapshot nature of the available satellite observations [Gryspeerd et al., 2015].*"

p.4, line 1: presumably part of the reason for choosing these regions is also their significance in terms of global convective activity?

Yes. we added "*...though it is possible that meteorological factors may also be of importance.*"

p.5, line 13: in addition to aerosol lifted by convection, the prior altitude of the aerosol layer in the environment can play an important role especially for absorbing aerosol. Given the study is based on CALIOP aerosol data, which has good vertical resolution, it is a shame that the analysis doesn't explicitly consider the different responses due to aerosol at different levels.

The collocation of aerosol and cloud data is conducted on a profile by profile basis, which considers the aerosol and cloud at all vertical levels. It is true that the aerosol environment before or after the formation of a cloud is unknown since CloudSat/CALIPSO observations are simply snapshots.

p.6, lines 2–3: "is consistent with the aerosol invigoration effect" is true, but it is important to acknowledge that this in no way demonstrates that this effect is in fact the mechanism.

Agree, we added "...*though it is possible that meteorological factors may also be of importance.*"

p.6, lines 10–11: the suggestion that the pollution is dimming enough to weaken the convection is highly speculative without evidence to demonstrate the mechanism.

Agree, we changed "suggests" to "*might suggest*".

p.6, line 18: the environmental changes are not restricted to moisture, but also mesoscale dynamics etc.

Agree, the original "i.e." was a typo, it should be "e.g.". Also, we added "... *changes in mesoscale dynamics*" as another example.

p.6, lines 20–21: changes due to large-scale conditions may also be the result of large-scale aerosol–radiation and aerosol–cloud interactions.

Agree. We have modified the paragraph there to include this suggestion.

p.7, lines 8–9: the dominance of increased CAPE doesn't in itself explain the reversed smoke effect – it would only explain a reduced or absent effect.

Agree. We have modified the paragraph there to include this suggestion.

p.7, lines 27, 30: again, it is not demonstrated that these mechanisms (unconsumed CAPE, decreased sunlight) are in fact the cause of the effects seen.

The mechanism about reduction in depletion of convective energy by absorbing aerosol was discussed by Wang et al. [2013] and recently supported by observations [Guo et al., 2016] and convection-resolving simulations [Lin, et al., Lee et al., 2016]. The latter two references have been added now. The surface dimming effect of absorbing aerosols are well documented in many previous literature.

Added references:

Guo, J., M. Deng, S.S. Lee, F. Wang, Z. Li, P. Zhai, H. Liu, W. Lv, W. Yao, X. Li, Delaying precipitation and lightning by air pollution over the Pearl River Delta. Part I: observational analyses, *J. Geophys. Res. Atmos.*, 121 (2016), 6472-6488.

Lin Lin Y., Y. Wang, B. W. Pan, J. X. Hu, Y. G. Liu, and R. Y. Zhang (2016), Distinct impacts of aerosols on an evolving continental cloud complex during the RACORO field campaign. *J. Atmos. Sci.*,73(9),3681-3700,https://doi.org/10.1175/jas-d-15-0361.1.

Lee, S.S., J. Guo, Z. Li, Delaying precipitation by air pollution over the Pearl River Delta: 2. Model simulations, *J. Geophys. Res. Atmos.*, 121 (11) (2016), 739-11,760

p.7, lines 31–32: can it be shown whether the different dust behaviours are due to differences in the dust itself, the convective regimes in question, or other external factors?

It is possible that the differences could be caused by the difference in the dust itself. Depending on the properties of the minerals that comprise the dust grains, they have could absorb lesser or greater amounts of light. Generally, biomass smoke and desert dust are known to absorb light. A further complication is the altitude of the dust. We have revised the text: *“The dust aerosols behave more like smoke in Southeast Asia and Central Africa. We note that dusts do not originate over South America – they originate from Africa. The Southeast Asia and Central Africa dusts originate from the same continent. The South American dust likely has a different altitude profile form the Southeast Asia and Central Africa dust. Dust at lower altitudes influences the temperature profile and atmospheric stability in a manner different from dust at higher altitudes.”*

p.8, lines 10–11: "more sensitive to aerosol type, not just RH" suggests that aerosol type has a larger impact on convection than RH, which seems unlikely and is probably not the intended meaning. Please clarify.

Now we have revised the sentence as *“our study finds the cloud heights are sensitive to aerosol type, regardless of moisture content in the atmosphere”*.

p.8, line 19: different-signed effects are also still seen for the same aerosol type in some cases between regions, seasons and RH/CAPE bins, so these are not fully disentangled.

We have revised the manuscript and this part has now been rewritten.

p.8, line 22: it is not just the individual cloud lifecycle, but also that of cloud systems and the mesoscale that may be relevant here.

Agree. Now we have revised the statement as *“For example, polar-orbit satellite data, measured at a fixed 1:30 PM ascending node observation time, only provide a transient relationship between aerosol and clouds, and do not address aerosol effects on cloud lifecycle and time-dependent mesoscale convection system issues.”*

p.9, lines 25–28: what kind of data is ECMWF–AUX? Is this pure model, pure observations, operational analysis, reanalysis (e.g. ERA-Interim) or something else?

The ECMWF–AUX is an operational analysis dataset that contains the set of ancillary ECMWF state variable data interpolated to each CloudSat cloud profiling radar bin.

p.10, line 2: the exclusion of aerosol below 500m risks excluding **exactly** the near-surface boundary-layer aerosol which is being entrained into the convective cloud base and thus driving any effects based on CCN activity. This is likely to be especially true in in the context of convection out of a shallow but heavily-polluted boundary layer. The potential impacts on the results should be discussed.

500 m is already inside the PBL, especially during the day time. Aerosols are typically well mixed by turbulence within PBL. More importantly, the aerosols near the upper part of the PBL (above 500 m) are more closely linked to the CCN near the cloud bottom (typically 1-2 km for deep convective clouds). Therefore, lack of aerosol information below 500 m should not be a concern for the ACI analyses in this and all other previous studies using CALIPSO. Also, CloudSat cannot detect clouds lower than 1 km, thus no clouds in our analysis will have a base lower than 500m.

p.10, lines 22–24: this mentions computer code used which is available from the authors; however the policy checklist states that no custom code is involved. This should be clarified and any additional policy documentation completed for the code if necessary.

The code is written by our researchers for processing and analyzing the CloudSat and CALIPSO datasets according to the methodology described in the paper. Although we are happy to share the code upon request, the code is not produced for the purpose of public usage.

p.13, lines 24–25 and 32–33: these seasonal variations in dust may be correlated with seasonal variations in convection for reasons unrelated to aerosol–convection interactions (e.g. large-scale circulation patterns).

Agree.

p.15, lines 9–10: the text notes the possible combination of aerosol and meteorological changes, but the potential impacts of such non-causal covariations on the results needs to be considered in much more detail as discussed above.

We have revised the manuscript and this part has now been rewritten.

pp.16–17, Figs S4–S7: the qualitative shapes of these curves in most cases do **not** look consistent between low/high RH or CAPE cases. A clearer explanation of the particular features which **are** consistent amongst the many differences is needed.

We have revised the manuscript and this part is no longer exist.

Reviewer #2 (Remarks to the Author):

This is an appealing paper that pulls together and seems to make sense of several previous studies of the effect of changes in aerosol on deep convection. In many ways it's a fairly straightforward paper that essentially presents a slightly more in-depth analysis of satellite data used by other studies, but makes the critical step of accounting for the aerosol type. It's mostly a satellite analysis with quite a lot of speculation about what might cause the different responses to different aerosol types and loadings. The speculations make sense (and are broadly supported by other studies), but the theoretical understanding and interpretation of the results is a bit unsatisfying.

My main concern is that the meteorological environment is not very thoroughly explored as a potential compounding variable. The analysis is performed for two RH and CAPE ranges. However, I'm not completely convinced that the authors properly account for the possibility that aerosol loading can correlate with CAPE, CIN, RH, etc. To be convincing **I suggest the authors show a scatterplot of aerosols versus the meteorological factors that could influence the IWC centroid**. This is not the same thing as showing the same centroid v aerosol relationship at different RH. What does aerosol v RH look like when dcentroid/daerosol is positive and negative? This should be repeated for other factors too. To be honest I would be surprised if these relationships reversed at a particular aerosol load (sufficient to provide an alternative explanation of the data), but the study would be considerably strengthened if you carried out this additional analysis.

We thank the reviewer for the constructive suggestions. However, to thoroughly explore the possible influences of meteorological factors, instead of making each individual scatterplot, we decide to compute full and partial correlations between AOT and IWC centroid with and without

the possible influence of 12 meteorological factors, for three types of aerosols in four seasons over three regions. Details are provided on page 7 of the revised manuscript and in the Supplementary.

These new analyses results provide stronger support to our observational findings of the type-dependent aerosol effects on convective clouds. We believe that the inhibition effect of smoke and invigoration effect of polluted continental aerosol are generally robust, although meteorological conditions may play some limited role in certain region and season. Those findings are crucial to elucidate the role of atmospheric aerosols in the current climate change.

The discussion and comparison with previous studies on p8 is quite unsatisfying. I was left really not understanding why your study differed. In particular, the following statement needs to be properly supported: “In previous studies¹⁸, there are many contradictory findings (of opposite sign) in regard to aerosol-cloud effects. Our analyses help to disentangle this problem, since different signed aerosol-cloud effects are associated with different aerosol types.” I’d like to see more than “helps to disentangle”. You should point out exactly how each study reached an apparently contradictory result that can now be understood in the light of different aerosol types. There are not that many. Perhaps a table in the SI would be a good way of showing this case by case information.

We have rewritten this part to clarify the value of this study: “*Previous studies show aerosol can either suppress [Koren et al., 2008; Fan et al., 2008; Wu et al., 2011] or invigorate [Storer et al., 2014; Wang et al., 2014; Fan et al., 2018] the development of deep convective clouds. Our analyses highlight that, in addition to the environmental factors (e.g. moisture, wind shear, large-scale circulation considerations), the different aerosol types are one of the key factors to determine the sign of aerosol effects on convective clouds.*”

The appealingly contradictory findings about aerosol – deep convective cloud (DCC) interactions from each study have been discussed in the revised first and second paragraphs in the Introduction part. The complexity of mechanisms in aerosol–DCC interactions from each case study have been extensively reviewed and summarized in recent review articles [Tao et al. 2012; Altaratz et al. 2014; Rosenfeld et al. 2014; Fan et al., 2016], so we will not repeat this effort in this short paper.

Added References:

- Fan J. E., R. Y. Zhang, W.-K. Tao, and K. I. Mohr (2008), Effects of aerosol optical properties on deep convective clouds and radiative forcing, *J. Geophys. Res.*, 113, D08209, <https://doi.org/10.1029/2007JD009257>.
- Fan J, Rosenfeld D, Zhang Y, Giangrande S E, Li Z, Machado L A, Martin S T, Yang Y, Wang J and Artaxo P, (2018), Substantial convection and precipitation enhancements by ultrafine aerosol particles *Science*, 359 411–418.
- Storer, R. L., S. C. van den Heever, and T. S. L’Ecuyer (2014), Observations of aerosol-induced convective invigoration in the tropical east Atlantic, *J. Geophys. Res. Atmos.*, 119, 3963–3975, doi:10.1002/2013JD020272.
- Wu, L., H. Su, and J. Jiang (2011), Regional simulations of deep convection and biomass burning over South America: 2. Biomass burning aerosol effects on clouds and precipitation, *J. Geophys. Res.*, 116, D17209, doi:10.1029/2011JD016106.

Altaratz, Orit & Koren, Ilan & Remer, Lorraine & Hirsch, E. (2014). Review: Cloud invigoration by aerosols—Coupling between microphysics and dynamics. *Atmospheric Research*. s 140–141. 38–60. [10.1016/j.atmosres.2014.01.009](https://doi.org/10.1016/j.atmosres.2014.01.009).

Rosenfeld, D., et al. (2014), Global observations of aerosol-cloud-precipitation-climate interactions, *Rev. Geophys.*, 52, 750–808, doi:10.1002/2013RG000441.

There are quite a few typos and poor uses of English that I guess can easily be taken care of. There are a couple of relevant papers that could be cited if space allows:

Wall et al. (2014) highlighted the importance of environment in controlling the response to aerosols: <https://doi.org/10.1175/JAS-D-13-0158.1>

Cui et al. (2011) showed the non-monotonic effect of increases in aerosols for a range of cloud types, CAPE etc: <https://doi.org/10.5194/acp-11-3495-2011>

Thank you for pointing this out. Co-Authors Steven Massie and Ali Omar are native English speakers. They have carefully checked English phrasing in the revised paper. We also added the above two references: Wall et al. (2014) and Cui et al. (2011)

Reviewer #3 (Remarks to the Author):

Do Aerosols Inhibit or Invigorate Convection? A Tale of Three Aerosol Types

By, Jonathan H. Jiang¹, Hui Su¹, Lei Huang², Yuan Wang³, Steven Massie⁴, Bin Zhao², Ali Omar⁵, Zhien Wang⁶

General Comment

The analysis presented here provides some new lines of evidence for the complex aerosol responses in convective clouds observed in many previous studies. Specifically, this study reveals how the sign of the convective invigoration response can be influenced by aerosol type. The authors' draw on well-established methodology and state-of-the-art satellite datasets. While the CALIOP retrieval of aerosol type may be imperfect (and further discussion of this is warranted) the analysis nicely demonstrates that these opposing responses are robust through the in-depth analysis of meteorological regimes (CAPE and relative humidity) provided in the supplementary section. Overall, the paper is well written and the presented arguments are supported by the figures and conclusions.

Other comments

How reliable is the CALIOP aerosol type detection? The results of this study largely hinge on the lidar's ability to classify aerosol correctly. I therefore would have expected more discussion on this point. Several studies examine the aerosol type retrieval from CALIOP using a variety of sources of validation. For example, the Cape Verde region aerosol type is reliably classified (Tesche et al. 2013). However, Wu et al. (2014) results show that the detection of very dense smoke layers are sometimes classified as cloud and thin aerosol layers are not commonly detected. Kacenelenbogen et al. (2014), JGR compare CALIOP with the High Spectral Resolution Lidar (HSRL) flown on the NASA Langley aircraft in coincident locations as CALIOP. Kacenelenbogen et al. (2014) demonstrates that the CALIOP retrieved aerosol type is actually fairly different from the one derived from HSRL and CALIOP tends to mostly misclassify smoke, polluted continental, dust, and clean marine aerosols. Mielonen et al. (2009) shows that CALIOP agrees with

AERONET aerosol type about 70% of the time. Aerosol type from CALIOP has also been compared to AeroCom models showing weak sensitivity to the extinction mean height diagnostic in industry and maritime locations but high sensitivity in African and Chinese dust regions (Koffi et al. 2012). From this literature review it is clearly difficult to determine exactly the accuracy of the CALIOP aerosol type retrieval from these studies. Broadly speaking, this is a challenging issue. Nonetheless, I would recommend including a few sentences on the validation of CALIOP aerosol type so the reader understands more of the sources of uncertainty. I am also sure whether there may be potential issues with the aerosol typing beneath relatively thick cirrus or cloud layers but this should also be investigated to ensure the retrievals are suitable in these locations too.

The reviewer correctly points out that the study relies on the lidar's ability to correctly classify aerosol types. Our study uses the new CALIPSO Level 2 Version 4 products which address most of the concerns. The new Level 2 Version 4 aerosol data products, released in November 2016, include substantial improvements to the aerosol subtyping and lidar ratio selection algorithms. These improvements likely mitigate the issues raised by the reviewer in papers that used the prior Version 3 data. We use Version 4 in our analysis to quantify how aerosols inhibit or invigorate convection.

To address the issues individually:

Wu et al (2014) raises two issues- (1) Some very dense smoke layers are classified as clouds and (2) thin aerosols are not always detected:

- (1) In version 3, any optically thick layer, defined as a layer that can be identified at single shot resolution (1/3 km resolution) was automatically deemed to be a cloud layer. In version 4, optically thick layers, including those found at the single shot resolution are classified using the cloud aerosol discrimination (CAD) algorithm. The aerosols found are classified using the subtyping algorithm. This solves most of the misclassification of dense aerosol layers as clouds.
- (2) The failure of CALIPSO to detect certain thin aerosols is an instrument limitation that depends on the signal-to noise ratio and more fundamentally on the laser energy and is therefore an inherently instrument design limitation that cannot be mitigated. However, the aerosol detection thresholds are discussed and quantified in Kim et al. (2018) and should be taken into account when analyzing the data from this paper.

Kacelenebogen et al. (2014) and Mielonen et al. (2009) raise some issues with the typing accuracy. CALIPSO aerosol classification has been extensively studied by comparisons with HSRL (Kacelenebogen et al. (2014) and Burton et al. (2013)). CALIPSO aerosol classification by Burton et al. [2013] shows relatively poor agreement for polluted dust (i.e., 35% of CALIPSO agrees with the HSRL-1 results) and smoke (13%), compared to marine (62%), polluted continental (54%), and desert dust (80%). In Version 4, there are major changes to the polluted dust typing scheme. Classification of the PD type is improved by introducing a new "dusty marine" aerosol subtype representing mixtures of dust and marine aerosols near the ocean surface which were previously misclassified as PD. The definitions of the elevated aerosol altitudes have been corrected leading to improvements in both the clean marine and smoke detections in Version 4. It is therefore safe to assert that algorithm enhancements in Version 4 chronicled in Kim et al. (2018) have addressed many of the issues raised by Kacelenebogen et al. (2014), Burton et al. (2013), and Mielonen et al. (2009).

Koffi et al. (2012) raises issues with sensitivity to the mean extinction height metric but also correctly point out that potential CALIOP and model limitations, and methodological factors might be responsible. In particular the AeroCom studies were for year 2000 while the CALIPSO measurements were for 2006-2009. It is not clear how the Version 4 CALIPSO products will affect the Koffi et al (2012) results.

In general the Version 4 updates result in a global mean 532 nm AOD retrieved by CALIPSO has increased from 0.084 to 0.128 (0.090 to 0.126) for nighttime (daytime). Lidar ratio revisions are the most influential factor for AOD changes from V3 to V4, especially for clear skies. Preliminary validation studies show that the AOD discrepancies between CALIPSO and MODIS(ocean)/AERONET are reduced in Version 4 compared to Version 3. In summary, the Version 4 Level 2 products used in these analyses, address a majority of the aerosol typing concerns raised in the validation of version 3 products.

References:

- Kim, M.-H., A. H. Omar, M. A. Vaughan, D. M. Winker, C. R. Trepte, Y. Hu, Z. Liu, and S.-W. Kim (2017), Quantifying the low bias of CALIPSO's column aerosol optical depth due to undetected aerosol layers, *Journal of Geophysical Research: Atmospheres*, n/a-n/a.
- Man-Hae Kim, Ali H. Omar, Jason L. Tackett, Mark A. Vaughan, David M. Winker, Charles R. Trepte, Yongxiang Hu, Zhaoyan Liu, Lamont R. Poole, Michael C. Pitts, Jayanta Kar, Brian E. Magill (2018) The CALIPSO Version 4 Automated Aerosol Classification and Lidar Ratio Selection Algorithm, in final preparation for submission to ACP.
- Burton, S., R. Ferrare, M. Vaughan, A. Omar, R. Rogers, C. Hostetler, and J. Hair (2013), Aerosol classification from airborne HSRL and comparisons with the CALIPSO vertical feature mask, *Atmospheric Measurement Techniques*, 6(5), 1397-1412.
- Koffi, B., M. Schulz, F.-M. Bréon, J. Griesfeller, D.M.M. Winker, Y. Balkanski, S. Bauer, T. Berntsen, M. Chin, W.D. Collins, F. Dentener, T. Diehl, R.C. Easter, S.J. Ghan, P.A. Ginoux, S. Gong, L.W. Horowitz, T. Iversen, A. Kirkevag, D.M. Koch, M. Krol, G. Myhre, P. Stier, and T. Takemura, 2012: Application of the CALIOP Layer Product to evaluate the vertical distribution of aerosols estimated by global models: Part 1. AeroCom phase I results. *J. Geophys. Res.*, 117, no. D10, D10201, doi:10.1029/2011JD016858.

Pg1 L21: “regional” to “regionally”

Corrected.

Pg2 L21: May want to mention that ultrafine aerosol particles also have recently been shown to enhance convection (Fan et al. 2018).

This reference has been added.

Pg3 – L4-5 and L9: Other studies have also sorted responses by cloud types. For example, Christensen et al. (2016) used the same CloudSat (CLDCLASS-LIDAR) data to quantify aerosol indirect radiative effects for deep convective clouds. Similarly, cloud types retrieved from MODIS were used in Oreopoulos et al. (2017). These studies also found non-linear aerosol responses of

the deep convective clouds (for example as you discuss on pg 17 L8) but did not sort by aerosol type as was uniquely conducted in your study.

Thanks for the helpful comments. Discussion of above studies has been added.

Pg3 L24-27: I would like to point out another study which uses a similar method to quantify the shift in the altitude centroid of deep convective clouds. Storer et al. (2014) used CloudSat to demonstrate that polluted clouds have a higher reflectivity centroid (center of mass concept) compared to unpolluted clouds over the central Atlantic Ocean.

We have added the Storer et al. (2014) reference when we discuss the mentioned about potential aerosol effect on convective invigoration.

Pg4 – L10: A pesky little point but I would replace the word “clear” with “obvious” or “evident” to avoid implying that “smoke” is actually “clear.”

We changed “clear” to “obvious”.

Figure 1: What is considered polluted versus unpolluted? The following paragraph suggests $AOT < 0.2$ is unpolluted but the supplementary materials suggest it is the case when “no aerosols (above 500m)” are detected {Pg10 L3-4}.

A polluted or aerosol environment is for cases with $AOT > 0$, while a clean case means $AOT = 0$ (no aerosol detected by CALIPSO above 0.5 km. CALIPSO cannot retrieve near surface aerosol below 0.5 km).

Pg6 L10: This sentence should probably specify which type, smoke or polluted continental aerosols, is being considered here. The previous sentence refers to heavy smoke aerosols. I assume this sentence is referring to polluted continental aerosols although it is not specifically stated.

We have added text to specify this is when the polluted continental aerosol dominate.

Pg10 L3-4: Can the definition of “clean” be clarified? Is it possible that “no aerosols” will be detected above 500 m if the CALIOP lidar is fully attenuated by an overlying cirrus cloud? In such a case, I could imagine the air being very “polluted” but these aerosols would be missed and classified as “clean” in the analysis simply because the lidar has no sensitivity due to attenuation. How are these cases treated?

We added “(i.e. no aerosols)” after “clean” to clarify that this refers to “no aerosols”. Yes, it is true that aerosols directly underneath the clouds cannot be detected. A major limit of active remote sensing is that it is extremely difficult to observe aerosols when the aerosols are inside or below the clouds. An effective way to deal with this observational difficulty is to analyze an extended record of satellite measurements in a statistical sense. CALIPSO and CloudSat provide daily global observations with fine resolution of 2-3 km horizontally. The determination of whether or not a cloud is contaminated by aerosols is determined by examining the areas near-by the clouds, not directly below the clouds.

Figure S2: It would be noteworthy to point out that Fig S2c is the same as Figure 1, just shown on different scales.

Yes, we added this note in the figure caption.

Reviewer #1 (Remarks to the Author):

This manuscript presents a new attempt to disentangle the various competing aerosol effects on convection based on satellite observations, using active sensing from the CALIPSO and CloudSat platforms to discriminate between aerosol types and to probe cloud phase and vertical structure. Aerosol–convection interactions are some of the most uncertain aspects of the climate impact of aerosol, so any improvement in understanding may have significant implications. The analysis focusses on the correlation of column-integrated aerosol optical thickness (AOT) with the vertical centroid of ice mass (Z_{IWC}). Different relationships are shown for different aerosol types, with most cases showing nonlinear (and sometimes non-monotonic) responses to increasing AOT.

While the attribution of causality in the relationships which are demonstrated remains a tricky issue as in many observation-based studies, the revised manuscript has substantially addressed most of the reviewers' concerns, with the additional table of partial correlations to exclude meteorological co-variations being particularly welcome, and presenting a more balanced picture of the conclusions. This is a well-written manuscript delivering an informative and somewhat novel analysis, and in its revised form I am now pleased to recommend its acceptable subject to minor revisions:

p.8, lines 4–5 and Table 1: closer consideration could be given to the reasons why one region (Central Africa) shows little statistically significant aerosol effect once the meteorological effects are removed, while in the other two regions the effect appears quite robust. (Table S1 suggests that a larger role of seasonality might be key here, with meteorology dominating on the annual timescale, but aerosol effects still manifesting on the intra-seasonal timescale.) This should be explored in the discussion.

In addition, a couple of minor typographical corrections:

p.2, line 10: "by the absorbing aerosol" -> "by absorbing aerosol".

p.10, line 1: "displays" -> "display".

Reviewer #2 (Remarks to the Author):

The authors have done a good job of analysing the meteorological covariability question. I hope they found my comment useful, because it clearly shows that aerosol effects are likely to be dominant in fewer regions and seasons than was originally stated. The additional analysis enhances the paper a lot.

However, I don't think the authors make enough of the analysis. The revised ms reads as if the authors are paying lip service to it, rather than really exploiting what it is telling them. In particular the added text on p8 is quite cursory. Then in the conclusions you say only that "in addition to environmental factors...aerosol type is one of the key factors..." This is still quite vague. The meteorological analysis wipes out any effect in central Africa. In some places meteorology works in the same direction as aerosol and in other places it works in the opposite direction. So in some places you can be even more confident of the aerosol effect, but on other places less confident. I suggest you explain in a bit more detail what the meteorological analysis tells us. And the caveats should be brought through to the conclusions (and probably the abstract) in a quantitative rather than qualitative way.

Reviewer #3 (Remarks to the Author):

I thank the authors for their clear and descriptive responses to my review and for adding the supporting details on the aerosol type retrieval from CALIPSO. Furthermore, I commend the authors for their in-depth analysis of examining the aerosol-cloud responses over a wide range of meteorological variables (as these aspects were prompted by the first two reviewers). In general, I tend to agree with the interpretation of the results and conclusions of the paper and now feel more confident in the analysis. Overall, I think this work provides a step change in our understanding of the influence of aerosol type in convective aerosol-cloud interactions. I do not have anything further to address at this time.

Reviewers' comments:

Reviewer #1 (Remarks to the Author):

This manuscript presents a new attempt to disentangle the various competing aerosol effects on convection based on satellite observations, using active sensing from the CALIPSO and CloudSat platforms to discriminate between aerosol types and to probe cloud phase and vertical structure. Aerosol–convection interactions are some of the most uncertain aspects of the climate impact of aerosol, so any improvement in understanding may have significant implications. The analysis focusses on the correlation of column-integrated aerosol optical thickness (AOT) with the vertical centroid of ice mass (ZIWC). Different relationships are shown for different aerosol types, with most cases showing nonlinear (and sometimes non-monotonic) responses to increasing AOT.

While the attribution of causality in the relationships which are demonstrated remains a tricky issue as in many observation-based studies, the revised manuscript has substantially addressed most of the reviewers' concerns, with the additional table of partial correlations to exclude meteorological co-variations being particularly welcome, and presenting a more balanced picture of the conclusions. This is a well-written manuscript delivering an informative and somewhat novel analysis, and in its revised form I am now pleased to recommend its acceptable subject to minor revisions:

p.8, lines 4–5 and Table 1: closer consideration could be given to the reasons why one region (Central Africa) shows little statistically significant aerosol effect once the meteorological effects are removed, while in the other two regions the effect appears quite robust. (Table S1 suggests that a larger role of seasonality might be key here, with meteorology dominating on the annual timescale, but aerosol effects still manifesting on the intra-seasonal timescale.) This should be explored in the discussion.

The reviewer is correct that in Central Africa the annual mean impact of aerosols seem insignificant, as indicated by the insignificant correlation coefficients. A closer look at the aerosol effects and meteorological influence in different seasons of Central Africa reveals the existence of the contrasting aerosol effects over the different time periods of a year. Both smoke and continental pollution aerosols show a significantly suppressing effect on convection during winter time, but their signs are reversed in the fall season, resulting in a net insignificant aerosol effect in the annual mean. A comparison of Table 1 and Table S1 overall suggests that seasonality in meteorology plays a role in Central Africa, with aerosol effects manifesting on the seasonal timescale, but meteorology dominating on the annual mean. We have added discussions of this in the subsection “Influence of meteorological factors” of the revised manuscript.

In addition, a couple of minor typographical corrections:

p.2, line 10: "by the absorbing aerosol" -> "by absorbing aerosol".

Corrected, thanks!

p.10, line 1: "displays" -> "display".

Corrected, thanks!

Reviewer #2 (Remarks to the Author):

The authors have done a good job of analysing the meteorological covariability question. I hope they found my comment useful, because it clearly shows that aerosol effects are likely to be dominant in fewer regions and seasons than was originally stated. The additional analysis enhances the paper a lot.

However, I don't think the authors make enough of the analysis. The revised ms reads as if the authors are paying lip service to it, rather than really exploiting what it is telling them. In particular the added text on p8 is quite cursory. Then in the conclusions you say only that "in addition to environmental factors...aerosol type is one of the key factors..." This is still quite vague. The meteorological analysis wipes out any effect in central Africa. In some places meteorology works in the same direction as aerosol and in other places it works in the opposite direction. So in some places you can be even more confident of the aerosol effect, but on other places less confident. I suggest you explain in a bit more detail what the meteorological analysis tells us. And the caveats should be brought through to the conclusions (and probably the abstract) in a quantitative rather than qualitative way.

We thank the reviewer for the constructive comments. We agree that not all the factors are explainable without future detailed modeling studies. However, the bulk behaviors of the inhibition effect of smoke and the invigoration effect of polluted continental aerosol shown in our study are robust no matter what meteorological factors are considered. This is especially true for the smoke aerosol. We have revised discussions in the subsection "Influence of meteorological factors".

Also, the main caveat using CloudSat and CALIPSO data in the study is the fact that they are from polar-orbit satellites, measured at fixed 1:30 AM/PM observation times. These data only provide an instantaneous relationship between aerosol and clouds, and miss the aerosol effects on cloud lifecycle and time-dependent mesoscale convection systems. We have discussed this caveat in the "Discussion" section.

Reviewer #3 (Remarks to the Author):

I thank the authors for their clear and descriptive responses to my review and for adding the supporting details on the aerosol type retrieval from CALIPSO. Furthermore, I commend the authors for their in-depth analysis of examining the aerosol-cloud responses over a wide range of meteorological variables (as these aspects were prompted by the first two reviewers). In general, I tend to agree with the interpretation of the results and conclusions of the paper and now feel more confident in the analysis. Overall, I think this work provides a step change in our understanding of the influence of aerosol type in convective aerosol-cloud interactions. I do not have anything further to address at this time.

Thank you!